# Risk of hepatitis B virus reactivation in the treatment of HBsAg and HBV DNA double-negative lymphoma patients

Lan Chen[1], Zihuan Lu[2], Xinqiang Zhang[2], Jinxin Lai[2], Ge Huang[2], Youwei Zheng 🔴[2]*

1 Department of Laboratory Medicine, Guangdong Provincial People's Hospital Ganzhou Hospital, Ganzhou Municipal Hospital, Ganzhou, China, 2 Laboratory Medicine, Guangdong Provincial People's Hospital (Guangdong Academy of Medical Sciences), Southern Medical University, Guangzhou, Guangdong Province, China

* zhengyouwei@gdph.org.cn

## Abstract

### Background

It is well-known that lymphoma patients undergoing treatment are at risk of hepatitis B virus (HBV) reactivation. This study aims to explore the risk factors for HBV reactivation in lymphoma patients who tested negative for both hepatitis B surface antigen (HBsAg) and HBV DNA before treatment, during their course of therapy. It provides clinical evidence for early intervention in HBV reactivation and rational preventive antiviral treatment.

### Methods

From January 2019 to December 2021, a total of 1,229 patients were diagnosed with lymphoma at Guangdong Provincial People's Hospital. Among them, 616 patients who tested negative for both HBsAg and HBV DNA and underwent therapy were recruited for the study. The recruited patients had a median age of 53.9 years (range: 14–88 years), with 358 males (58.12%) and 258 females (41.88%). The risk factors associated with HBV reactivation in these patients were then analyzed.

### Results

Among the 616 lymphoma patients enrolled in this study, 44 patients (7.14%, 44/616) exhibited HBV reactivation. Notably, the rate of HBV reactivation was significantly higher in patients with hepatitis B core antibody (HBcAb) (+) (10.00%) compared to those with HBcAb (-) (1.46%) (P < 0.001, OR = 7.52). An analysis of HBV reactivation rates across different age groups demonstrated a statistically significant difference (P = 0.002). In particular, patients aged over 60 years showed a markedly elevated rate of HBV reactivation compared to those in other age brackets (P < 0.001). Conversely, no statistically significant differences in HBV reactivation rates were

**Data availability statement:** All relevant data are within the manuscript and its Supporting Information files.

**Funding: Funding:** The author(s) received no specific funding for this work.

**Competing interests:** The authors have declared that no competing interests exist.

observed between patients of different genders (P = 0.637, OR = 0.855) or across varying treatment durations (P = 0.851).

## Conclusion

For lymphoma patients undergoing treatment, HBV reactivation may occur even if both HBsAg and HBV DNA are negative at the initiation of treatment. It is noteworthy that this study found that patients with HBcAb (-) also experienced HBV reactivation. Therefore, patients who are negative for HBsAg, HBcAb, and HBV DNA should also be closely monitored to mitigate the risk of HBV reactivation.

---

## Introduction

Lymphoma, as a malignant tumor originating from the lymphohematopoietic system, continues to see a rising incidence rate globally, posing a serious threat to human life and health [1,2]. In the course of treating lymphoma, patients often need to undergo various treatment modalities such as chemotherapy, radiotherapy, or immunotherapy. However, while these treatment modalities effectively target and kill tumor cells, they may also cause varying degrees of damage to the patient's body, leading to a series of complications [3–6].

Hepatitis B virus (HBV) reactivation is a common complication in lymphoma patients undergoing treatment, potentially leading to serious consequences such as hepatitis, acute liver failure, and even life-threatening situations [7,8]. Generally speaking, reactivation is characterized by an increase in HBV DNA levels from baseline in chronic hepatitis B (CHB) patients, or the reappearance of hepatitis B surface antigen (HBsAg) in patients who were previously HBsAg (-). Additionally, HBV DNA may become detectable in the serum of patients who are both HBsAg and HBV DNA negative [9,10].

Previous studies have primarily focused on lymphoma patients who are HBsAg (+) or HBsAg (-)/hepatitis B core antibody (HBcAb) (+). HBsAg (+) patients exhibit active HBV replication due to detectable surface antigen in serum, resulting in an intuitive reactivation risk mechanism under immunosuppressive therapy, for which extensive research has established a theoretical basis for clinical prevention strategies [11,12]. In addition, HBsAg (-)/HBcAb (+) patients, whose positive HBcAb indicates past HBV infection with potential residual viral components persisting in hepatocytes, have also been documented to carry a reactivation risk during immunosuppressive therapy [13–15].

Despite limited research on HBV reactivation risk in HBsAg (-)/HBV DNA (-) patients, its clinical significance cannot be overlooked. The risk arises from two aspects: first, patients with initially undetectable serum HBsAg may harbor latent HBV infections due to persistent covalently closed circular DNA (cccDNA) in hepatocytes, which serves as the transcriptional template for viral replication [16–18]; second, as an important biomarker of HBV infection, the status of HBcAb also plays a role in HBV reactivation [19,20]. At present, the lack of risk assessment guidelines for

this group in clinical practice results in inadequate reactivation monitoring, thereby compromising treatment efficacy and increasing mortality.

Given the profound impact of HBV reactivation on the prognosis of lymphoma patients and the significant research gap concerning HBsAg (-)/HBV DNA (-) lymphoma cases, this study aims to investigate risk factors for HBV reactivation during treatment, with particular emphasis on the influence of HBcAb status. By addressing this knowledge gap, we aspire to establish more precise risk assessment metrics for clinical practice, optimize preventive strategies, and ultimately improve patient outcomes.

## Materials and methods

### Study population

This study retrospectively analyzed lymphoma patients admitted to the Lymphoma Department of Guangdong Provincial People's Hospital between January 1, 2019, and December 31, 2021. A total of 1,229 patients were initially screened, with the following inclusion criteria: (1) diagnosed with lymphoma; (2) negative serum HBsAg and HBV DNA at diagnosis; and (3) ability to undergo regular monitoring of HBsAg, HBcAb, and HBV DNA during treatment. Based on these criteria, 616 patients were ultimately included in the retrospective analysis (Fig 1).

### Methods

The observation starting point of this study is the initial enrollment time, with the endpoint being the occurrence of HBV reactivation in patients. In the absence of HBV reactivation, patients will be tracked and observed until December 31, 2022. This study has been approved by the Ethics Committee of Guangdong Provincial People's Hospital (Ethics Code: KY-N-2022-120-01).

**Monitoring frequency and detection methods for HBsAg, HBcAb, and HBV DNA.** During treatment, to accurately monitor HBV infection status and reactivation risk, HBsAg, HBcAb, and HBV DNA testing were strictly standardized. For each scheduled hospital visit, patients underwent simultaneous assessment of HBV reactivation potential, including these three biomarkers. Given that dynamic factors such as immune function, tumor burden, and treatment modality can influence HBV status during therapy, this frequent monitoring enabled timely detection of changes to guide treatment adjustments.

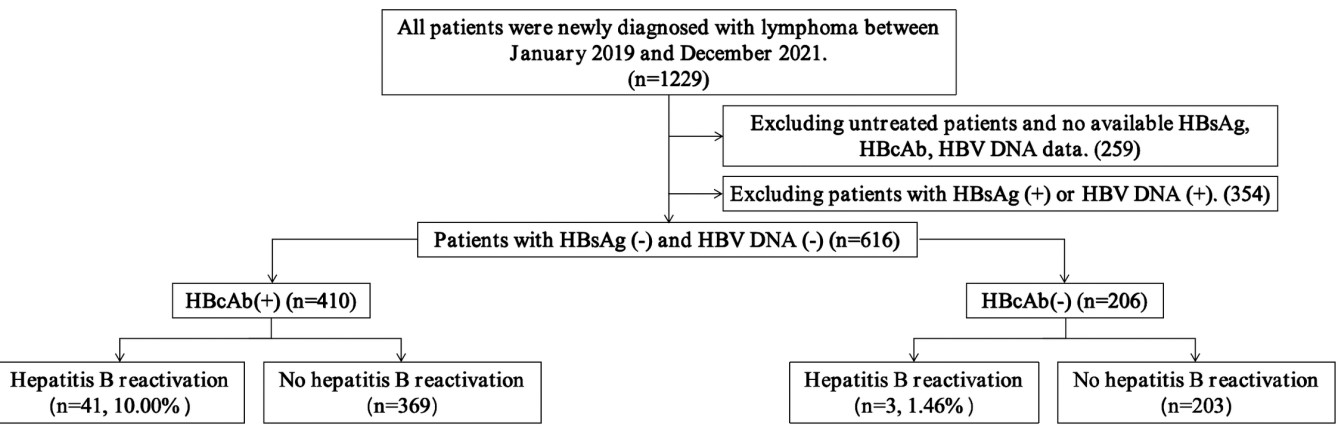

**Fig 1. Selection process for HBsAg and HBV DNA double-negative patients.**

Detection methods: Quantitative HBsAg and HBcAb assays were performed using the Cobas E602 fully automated electrochemiluminescence analyzer with dedicated reagent kits (Roche, Basel, Switzerland). HBV DNA quantification was conducted on the COBAS AmpliPrep/COBAS TaqMan system with dedicated reagent kits (Roche, Basel, Switzerland).

Positive criteria:

- HBsAg: Cut-off index (COI) > 1.000

- HBcAb: COI < 1.000

- HBV DNA: > 20.00 IU/mL (lower limit of quantification: 20.00 IU/mL)

Results exceeding these thresholds were considered positive.

**Diagnostic Criteria for HBV Reactivation.** HBV DNA may turn positive, or HBsAg may change from negative to positive [9,10].

### Statistical analysis

All data were processed using SPSS 26.0 software for statistical analysis. The count data, including HBcAb status, age group, gender, and duration of treatment, are presented as proportions (%). To compare the intergroup differences in HBV reactivation among various factors, we used the chi-square test or Fisher's exact test, with a P-value < 0.05 indicating statistical significance. In addition, we included odds ratios (ORs) to compare the relationship between HBcAb status, gender, and HBV reactivation. An OR > 1 that falls within the 95% confidence interval (CI) indicates statistical significance.

## Results

### Impact of HBcAb status on HBV reactivation

During the treatment period observed in this study, 44 of 616 lymphoma patients experienced HBV reactivation (44/616, 7.14%) (Table 1). Among them, there were 206 patients with HBcAb (-), and 3 cases showed HBV reactivation (3/206, 1.46%). Among 410 patients with HBcAb (+), 41 cases showed HBV reactivation (41/410, 10%). Comparison between HBcAb (-) and HBcAb (+) patients showed that the frequency of HBV reactivation was significantly higher in HBcAb (+) patients than in HBcAb (-) patients, with a statistically significant difference between the two groups (P < 0.001). The probability of HBV reactivation in HBcAb (+) patients is 7.52 times higher than that in HBcAb (-) (OR = 7.52).

Among the 3 patients with HBcAb (-) undergoing HBV reactivation, the degree of reactivation varied. Patient 1 experienced HBV reactivation after 9 months of treatment, and both HBsAg and HBV DNA were detected as positive. The concentration of HBV DNA was as high as 5.66E + 03 IU/mL, but HBcAb remained negative (Fig 2A). Patient 2 showed persistent HBcAb (+) after 2 months of treatment, but HBV DNA was not detected until 10 months later (Fig 2B). Patient 3 remained negative for HBsAg and HBcAb for over a year of treatment, until HBV DNA became positive after 21 months of treatment, at which point HBsAg and HBcAb remained negative (Fig 2C).

### Impact of age and gender on HBV reactivation

The frequency of HBV reactivation was compared among patients of different age groups (Table 1), and the difference was statistically significant (P = 0.002). Notably, among the 240 patients aged over 60 years, 29 experienced HBV reactivation (29/240, 12.08%). This frequency of HBV reactivation was significantly higher compared to that observed in patients from other age groups (P < 0.001).

Additionally, among the 358 male patients, 24 experienced HBV reactivation (24/358, 6.70%). And among the 258 female patients, 20 experienced HBV reactivation (20/258, 7.75%) (Table 1). Although the incidence of HBV reactivation

**Table 1. The results of different factors affecting HBV reactivation.**

| Factors | Classifications | The number of observations | HBV reactivation | | P-value |
|---|---|---|---|---|---|
| | | | Yes (%) | No (%) | |
| HBcAb | Positive | 410 | 41 (10.00) | 369 (90.00) | < 0.001 |
| | Negative | 206 | 3 (1.46) | 203 (98.54) | |
| Age (Y) | 0 < Y ≤ 20 | 18 | 0 (0.00) | 18 (100.00) | 0.002 |
| | 21 < Y ≤ 30 | 51 | 0 (0.00) | 51 (100.00) | |
| | 31 < Y ≤ 40 | 57 | 0 (0.00) | 57 (100.00) | |
| | 41 < Y ≤ 50 | 93 | 4 (4.30) | 89 (95.70) | |
| | 51 < Y ≤ 60 | 157 | 11 (7.01) | 146 (92.99) | |
| | 61 < Y ≤ 70 | 147 | 17 (11.56) | 130 (88.44) | |
| | > 70 | 93 | 12 (12.90) | 81 (87.10) | |
| Gender | Male | 358 | 24 (6.70) | 334 (93.30) | 0.637 |
| | Female | 258 | 20 (7.75) | 238 (92.25) | |
| Duration of treatment (M) | 0 < M ≤ 3 | 616 | 12 (1.95%) | 604 (98.05%) | 0.837 |
| | 3 < M ≤ 6 | 517 | 14 (2.71%) | 503(97.29%) | |
| | 6 < M ≤ 9 | 297 | 5 (1.68) | 292 (98.32) | |
| | 9 < M ≤ 12 | 173 | 2 (1.56) | 171 (98.84) | |
| | 12 < M ≤ 15 | 115 | 1 (0.87) | 114 (99.13) | |
| | 15 < M ≤ 18 | 88 | 3 (3.41) | 85 (96.59) | |
| | 18 < M ≤ 21 | 69 | 1 (1.45) | 68 (98.55) | |
| | 21 < M ≤ 24 | 56 | 1 (1.79) | 55 (98.21) | |
| | 24 < M ≤ 27 | 46 | 0 (0.00) | 46 (100.00) | |
| | 27 < M ≤ 30 | 31 | 1 (3.23) | 30 (96.77) | |
| | 30 < M ≤ 33 | 18 | 0 (0.00) | 18 (100.00) | |
| | 33 < M ≤ 36 | 8 | 2 (25.00) | 6 (75.00) | |
| | 36 < M ≤ 39 | 5 | 1 (20.00) | 4 (80.00) | |
| | 39 < M ≤ 42 | 4 | 1 (25.00) | 3 (75.00) | |
| | 42 < M ≤ 45 | 3 | 0 (0.00) | 3 (100.00) | |
| | 45 < M ≤ 48 | 2 | 0 (0.00) | 2 (100.00) | |

For HBcAb status, the OR of HBV reactivation in HBcAb (+) individuals relative to HBcAb (-) individuals is 7.52, with a 95% CI of 2.24 to 25.22. For gender factor, the OR is 0.855, with a 95% CI of 0.455 to 1.604. Among the 616 patients, the age range is from 14 to 88 years, with an average age of 53.9 years. M: Months, Y: Years.

was slightly higher in female patients than in male patients, there was no statistically significant difference between the two groups (P = 0.637).

## Impact of treatment duration on HBV reactivation

Since patients with a treatment duration exceeding 30 months in each group are fewer than 20, they are excluded from the intergroup comparison of treatment durations. And there was no statistically significant difference in the incidence of HBV reactivation among lymphoma patients who underwent different treatment durations (P = 0.851). Except for patients who did not experience HBV reactivation within the 24–27 months treatment period, HBV reactivation occurred in all other stages. Among those who experienced reactivation, the highest frequency of HBV reactivation (3.41%) was observed during the 15–18 months treatment period, although there were only 3 cases. In contrast, 13 cases (2.51%) of HBV reactivation occurred during the 3–6 months treatment period (Table 1).

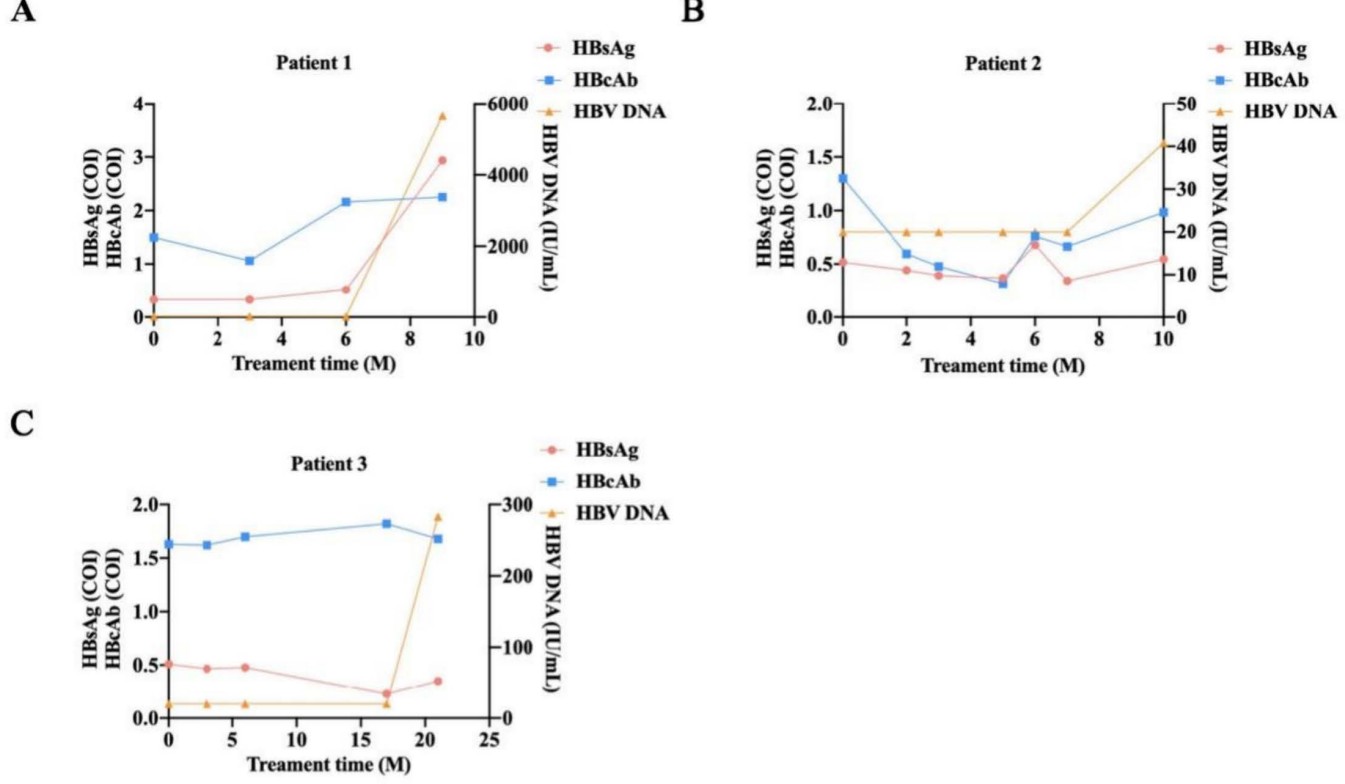

**Fig 2. Quantitative detection results of HBsAg, HBcAb, and HBV DNA in 3 lymphoma patients with HBcAb (-) who experienced HBV reactivation.** (In the figure, the detection results with < 20.00 IU/mL are uniformly set to a value of 20.00 IU/mL, indicating negative HBV DNA. M: Months.).

## Discussion

In this study, we investigated the risk factors for HBV reactivation in patients with HBsAg and HBV DNA double-negative lymphoma during treatment, with a particular focus on the impact of HBcAb status on the risk of HBV reactivation. Through the analysis of 616 patients, we discovered that even when both HBsAg and HBV DNA were negative during initial treatment, 7.14% (44 cases) of the patients still experienced HBV reactivation. This finding challenges the traditional notion that "HBsAg and HBV DNA double negative is safe" and indicates the necessity of reassessing monitoring strategies for such patients.

The current consensus is still to recommend serological screening for HBV infection before initiating chemotherapy or immunosuppressive therapy to assess the risk of HBV reactivation in patients. When the patient's HBsAg (+), HBV DNA<2.00E+03 IU/ml and ALT, AST levels are normal, they are classified as inactive carriers of hepatitis B virus and should receive preventive treatment for HBV reactivation. In China, the treatment plan for HBV infection mainly includes antiviral therapy, such as the use of nucleoside analogues (NAs) and other drugs. For patients with HBV DNA (-) and HBsAg (+) or HBsAg (-)/HBcAb (+), regular testing of ALT, AST, and HBV DNA levels is necessary to monitor for HBV reactivation [10,21]. However, the existing consensus primarily bases risk assessment strategies on the positive status of HBsAg and HBV DNA. Consequently, there remains a gap in risk stratification for patients who are double-negative for HBsAg and HBV DNA.

This study further reveals that HBV reactivation occurred in three HBcAb (-) patients despite the absence of detectable HBsAg and HBV DNA. Multiple mechanistic explanations may account for this phenomenon. On the one hand,

seronegative occult hepatitis B virus infection (OBI) represents a plausible etiology. In certain individuals, HBV may persist within hepatocytes in an extremely covert manner, with viral replication maintained at minimal levels insufficient to generate detectable HBsAg and HBcAb by conventional serological assays. Under this state of occult infection, cccDNA can establish long-term latency in hepatocytes, forming a stable viral reservoir. When patients receive immunosuppressive therapy, the diminished immune surveillance and control enable reactivation of the previously quiescent virus, leading to clinically apparent HBV reactivation [22,23]. On the other hand, immune escape variants may also represent a critical contributing factor. HBV demonstrates high mutability, enabling the emergence of immune escape mutant strains during chronic infection. These variants evade recognition and neutralization by host immune defenses, thereby impairing the generation of effective antiviral immune responses. Notably, even in individuals with documented prior HBV exposure, the presence of immune escape mutations may lead to false-negative results in conventional serological assays (e.g., HBsAg and HBcAb detection). Meanwhile, intrahepatic cccDNA persists as a stable viral reservoir, posing significant risks for viral reactivation under conditions of immunosuppression [24–26].

Additionally, rare cases of HBV reactivation have been documented in truly seronegative individuals (i.e., those testing negative for both HBsAg and HBcAb). Studies suggest that in specific clinical contexts—such as primary immunodeficiency or administration of intense immunosuppressive therapy—OBI may persist despite negative serological markers. This latent viral reservoir can undergo reactivation when immune homeostasis is disrupted, potentially triggering hepatitis and other liver-related complications [27]. These findings further underscore the clinical necessity for HBV monitoring in patients with negative HBsAg/HBcAb profiles.

Among three cases in our cohort showing HBV reactivation despite HBcAb (-) results, we considered multiple etiological possibilities. First, false-negative HBcAb results could not be excluded. While serological assays generally demonstrate high accuracy and sensitivity, procedural factors (e.g., specimen handling, reagent performance) may occasionally produce erroneous results. Second, delayed seroconversion might explain certain cases. During acute HBV infection, HBcAb production requires time, and individual variations in immune response kinetics may result in detectable intrahepatic cccDNA despite transiently negative serological markers at the time of testing. Third, low-level prior exposure below standard serological detection thresholds represents another plausible mechanism. Patients with mild or asymptomatic HBV infection may generate minimal HBcAb responses that remain undetectable by conventional assays, while residual intrahepatic cccDNA persists with potential for reactivation under immunosuppressive conditions.

Although nosocomial infection is a potential factor, based on our diagnostic criteria and strict infection control measures followed by the hospital, we believe the observed cases are primarily due to HBV reactivation. However, we recognize that in rare instances, nosocomial infection cannot be completely excluded. Future studies could further investigate this through more detailed tracking of infection sources.

In this study, the phenomenon of HBcAb conversion from negative to positive during treatment in some patients further supports this view. This transition may be due to virus replication stimulating the immune system to produce antibodies, or antibody levels that were originally below the detection limit being elevated to a measurable range as a result of changes in immune status. These findings indicate that HBcAb (-) is not an absolute "safe signal" and the risk of occult HBV infection still warrants vigilance.

The incidence of HBV reactivation in patients with HBcAb (+) was significantly higher than that in patients with HBcAb (-) (P<0.001). This is closely related to the pathological significance of HBcAb. A positive HBcAb result indicates that the patient has been infected with HBV. Even if HBsAg and HBV DNA are currently negative, there may still be residual cccDNA or integrated viral genomes in liver cells, which are more prone to reactivation under immunosuppressive conditions [22,28]. Furthermore, the risk of HBcAb (-) patients cannot be ignored. Although their frequency of reactivation is low, the mechanism is similar to that of HBcAb (+) patients. Moreover, factors such as the intensity of immune suppression and underlying liver disease may increase the risk of reactivation in these patients.

It is worth noting that age is also an independent risk factor. In this study, patients over 60 years old had a significantly higher frequency of HBV reactivation compared to patients in other age groups (P < 0.001). From an immunological perspective, this may be because immune function declines in elderly patients, weakening the immune system's ability to clear viruses. Meanwhile, this is also related to the reduced tolerance of liver cells to viruses. Under the influence of immunosuppressive therapy, the virus – host interaction in elderly patients, which was originally in a relatively balanced state, may be disrupted, leading to HBV replication and reactivation. These findings are consistent with previous research results [29–31], which indicate that both HBcAb (+) and elderly age are risk factors for HBV reactivation. In addition, we also considered factors such as gender and treatment duration. However, the results of this study showed that there was no statistically significant difference in the frequency of HBV reactivation among patients of different genders and treatment durations (P > 0.05).

This study is the first to explore the risk factors for HBV reactivation in lymphoma patients who are negative for both HBsAg and HBV DNA, filling a research gap in this field. Based on the study results, we believe that for lymphoma patients negative for both HBsAg and HBV DNA, it is necessary to regularly monitor HBV – related indicators regardless of whether they are HBcAb (+) or HBcAb (-), and closely watch for the possibility of HBV reactivation. Particularly for patients who are HBcAb (+) and of older age, greater emphasis should be placed on tracking HBV reactivation to enable timely detection and the initiation of antiviral therapy. However, this study solely focused on serological HBV reactivation and did not analyze the patients' clinical symptoms. In the future, further exploration of the association between serological indicators and clinical symptoms is required to provide a basis for developing precise prevention strategies.

## Supporting information

**S1 File. The specific values of HBsAg, HBcAb, and HBV DNA in** Fig 2.
(XLSX)

## Acknowledgments

We thank the Medical Record Room of Guangdong Provincial People's Hospital for providing the case database.

## Author contributions

**Data curation:** Zihuan Lu, Xinqiang Zhang.

**Formal analysis:** Lan Chen, Jinxin Lai.

**Investigation:** Zihuan Lu, Xinqiang Zhang.

**Methodology:** Jinxin Lai.

**Project administration:** Ge Huang, Youwei Zheng.

**Supervision:** Ge Huang, Youwei Zheng.

**Writing – original draft:** Lan Chen.

**Writing – review & editing:** Lan Chen, Youwei Zheng.

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
