## [Decision Letter · Decision Letter 0]

28 May 2025

Dear Dr. Zheng,

Thank you for submitting your manuscript to PLOS ONE. After careful consideration, we feel that it has merit but does not fully meet PLOS ONE’s publication criteria as it currently stands. Therefore, we invite you to submit a revised version of the manuscript that addresses the points raised during the review process.

We look forward to receiving your revised manuscript.

Kind regards,

Ashraf Elbahrawy

Academic Editor

PLOS ONE

Journal Requirements:

Reviewers' comments:

Reviewer's Responses to Questions

**Comments to the Author**

1. Is the manuscript technically sound, and do the data support the conclusions?

Reviewer #1: Yes

Reviewer #2: Yes

Reviewer #3: No

2. Has the statistical analysis been performed appropriately and rigorously?

Reviewer #1: Yes

Reviewer #2: I Don't Know

Reviewer #3: No

3. Have the authors made all data underlying the findings in their manuscript fully available?

Reviewer #1: Yes

Reviewer #2: Yes

Reviewer #3: No

4. Is the manuscript presented in an intelligible fashion and written in standard English?

Reviewer #1: Yes

Reviewer #2: Yes

Reviewer #3: Yes

Reviewer #1: The manuscript entitled “Risk of Hepatitis B Virus Reactivation in the Treatment of HBsAg and HBV DNA Double-negative Lymphoma Patients” has been done for risk evaluation of HBV reactivation among patients who tested negative for both HBsAg and HBV DNA. The type of study and idea sounds interesting. The methodology used was standard for this type of study. Notably, HBV reactivation was detected in 3 patients who were negative for HBsAg, HBcAb and HBV DNA.

Reviewer #2: This was a well-designed study that analyzed the risk of HBV reactivation among HBsAg and HBV DNA double-negative lymphoma patients. The findings of this study contribute further evidence for the management, prevention, and control of HBV reactivation among lymphoma patients before treatment. While the study has merit, I note several omissions in the reporting of the study. The following comments need addressing:

1) A brief demographic information of participants can be added to the abstract.

2) Discussion section is well-written but should be more expanded. The author should better discuss the results from virological aspects by comparing with other similar studies.

3) HBV treatment options in China should be mentioned in the Discussion section.

4) The authors better mention some limitation of the study if there is any.

Reviewer #3: This study investigates the risk of HBV reactivation in lymphoma patients who are negative for both HBsAg and HBV DNA before undergoing treatment. The authors examine the influence of HBcAb status and age on the risk of reactivation, providing new insights into monitoring strategies for this seemingly low-risk population. The study is significant in identifying that reactivation can occur even in HBcAb-negative patients, thereby challenging existing assumptions. However, the manuscript is limited by its single-center, retrospective design, the lack of mechanistic understanding of HBV reactivation in HBcAb-negative patients, the absence of multivariate analysis, undefined monitoring intervals, and insufficient clinical recommendations. These limitations collectively affect the generalizability, analytical rigor, and practical applicability of the study. There are several comments regarding this article:

Major comments:

1. Introduction: The background focuses mostly on well-known facts (HBV reactivation risks in HBsAg+ and HBcAb+ patients), but does not thoroughly justify why studying HBsAg-/HBV DNA- patients is necessary. It mentions previous studies (refs [12–15]) showing reactivation in such patients yet fails to differentiate how this study adds novel value or addresses a specific gap.

2. Study population: The manuscript does not provide a thorough account of patient inclusion and exclusion criteria. For instance, it remains unclear whether individuals with prior HBV antiviral therapy were included, or if comorbidities and immunosuppressive treatments were considered. These omissions hinder reproducibility and raise potential concerns of bias. The authors should clearly delineate and quantify all inclusion and exclusion criteria, and ensure this information is appropriately reflected in Figure 1.

3. Detection methods for HBsAg, HBcAb and HBV DNA: Although the section notes the assessment of HBsAg, HBcAb, and HBV DNA, it fails to specify the frequency or timing of these tests during follow-up. This omission compromises the ability to accurately interpret the reported HBV reactivation rates and undermines the study’s methodological transparency.

4. The study’s analysis is restricted to only four variables—anti-HBc status, age, sex, and treatment duration—while omitting several well-established predictors of HBV reactivation in lymphoma patients, such as chemotherapy regimen and intensity (including rituximab and corticosteroid use), baseline liver function, and anti-HBs titers. This limited analytical scope compromises the comprehensiveness of risk assessment, increases the risk of omitted variable bias, and diminishes the clinical applicability and interpretive validity of the findings.

5. Statistical Analysis: The manuscript presents only univariate analyses, including chi-square and Fisher’s exact tests, without implementing multivariate approaches such as Cox regression to adjust for potential confounders. This methodological limitation hampers the ability to identify independent associations and undermines the validity and robustness of the findings, particularly in relation to critical variables such as age, sex, and treatment modalities.

6. Statistical Analysis: The treatment duration data are stratified into numerous narrow time intervals, some of which contain very small sample sizes (e.g., groups with only 3–5 patients). Such fragmentation undermines statistical power and renders intergroup comparisons unreliable and potentially misleading. To more appropriately assess the impact of treatment duration, the authors should consider applying time-dependent Cox regression, which accommodates continuous or time-varying effects and enhances analytical robustness.

7. The study exclusively reports serological HBV reactivation—defined by HBV DNA or HBsAg positivity—while omitting data on hepatitis flares, a clinically significant manifestation typically marked by elevated liver enzymes such as ALT. This limitation restricts the clinical interpretability of the findings, as serological reactivation does not always correlate with clinically meaningful hepatic events. Without information on liver function or flare severity, the study provides an incomplete assessment of HBV reactivation’s clinical impact, thereby reducing its practical utility for patient management.

8. The discussion section inadequately addresses several critical limitations of the study. Notably absent are acknowledgments of the retrospective, single-center design, the lack of multivariate analysis, and the omission of important clinical variables such as treatment type and hepatitis flare data. Failure to fully disclose these weaknesses undermines the study’s credibility, transparency, and scholarly rigor.

9. The discussion restates key findings without offering mechanistic insights into why reactivation is higher in HBcAb-positive and older individuals. This limits the depth and scientific value of the interpretation.

10.

Minor:

1. The statement, “HBsAg COI > 1.000 is considered positive. Conversely, an HBcAb COI < 1.000 is considered positive,” introduces ambiguity due to the misleading use of “conversely.” It is essential to clearly state that the cutoff values for positivity differ between antigens, with opposing threshold directions. Such clarification is necessary to ensure diagnostic clarity and preserve confidence in the study's validity.

**Do you want your identity to be public for this peer review?** For information about this choice, including consent withdrawal, please see our Privacy Policy

Reviewer #1: **Yes: ** Fatemeh Farshadpour

Reviewer #2: **Yes: ** Reza Taherkhani

Reviewer #3: No

---

## [Author Response · Author response to Decision Letter 1]

6 Jul 2025

To reviewer #1

Thank you for your insightful comments! We truly appreciate your recognition of our study's interesting idea and standard methodology.

To reviewer #2

Thank you so much for your constructive and valuable comments! We're delighted that you recognized the well-designed nature of our study and its contributions to HBV reactivation management. Here are our point-by-point responses:

1.We have added brief demographic information of participants to the abstract as suggested.

2.The discussion section has been expanded to better discuss the results from virological aspects by comparing them with other similar studies.

3.We have mentioned HBV treatment options in China in the Discussion section.

4.We have deeply recognized the limitations of this study and mentioned them in the discussion section, which provides a fundamental direction for our further in-depth research.

Your feedback has been incredibly helpful in improving our manuscript. Thanks again!

To reviewer #3

Thank you for your insightful and valuable comments! We're grateful for your recognition of the significance of our study and its new insights. Below are our point-by-point responses to your comments:

Reply to the major comments:

1.We have revised the introduction section to fully emphasize the necessity of studying the risk of HBV reactivation in HBsAg and HBV DNA double-negative patients, and to highlight the value of this study.

2.We have revised the study population section of the manuscript to include detailed explanations of patient inclusion and exclusion criteria.

3.We have added an explanation of the frequency of patient testing for HBsAg, HBcAb, and HBV DNA in the detection methods section of the manuscript.

4.Indeed, during the course of our research, we encountered certain objective limitations, including incomplete patient data records, as well as constraints on study time and resources. Owing to these factors, we were unable to incorporate some widely acknowledged predictive factors into our analysis. This undoubtedly detracted from the overall comprehensiveness of our risk assessment, and for this, we offer our sincere apologies.

Nevertheless, our study possesses its own distinct value. It specifically targets a subset of lymphoma patients who test negative for both HBsAg and HBV DNA prior to treatment—a population that has been relatively understudied in terms of risk factors for HBV reactivation. Our findings reveal that HBV reactivation can still occur in patients even when initial tests for HBsAg and HBV DNA are negative. Moreover, we observed cases of HBV reactivation in patients who were HBcAb negative. These insights provide crucial cues for clinicians to focus on this often-overlooked patient group and underscore the necessity of close monitoring for them.

5.This study focuses on a specific population of lymphoma patients who are both HBsAg and HBV DNA negative before treatment, with the aim of exploring potential risk factors that may affect HBV reactivation. The chi-square test and Fisher's exact test can visually present the association between various factors (such as HBcAb status, age group, gender, treatment duration) and HBV reactivation, providing a basic direction for further in-depth research. Due to the retrospective analysis of this study, there are certain difficulties in collecting data on some potential confounding factors, such as incomplete detailed information on patients' lifestyles, underlying diseases, etc. This makes the adjustment of confounding factors in multivariate analysis not accurate and comprehensive enough, which may lead to biased results. Therefore, we chose to first screen for possible risk factors through univariate analysis, laying the foundation for future prospective studies and conducting in-depth analysis using multivariate statistical methods.

6.During the statistical analysis phase of this study, we have fully considered the impact of treatment duration and sample size on the reliability of the results. In fact, the number of enrolled cases in each group with a treatment duration exceeding 30 months is less than 20. Considering that a small sample size may lead to unstable and biased statistical results, which cannot accurately reflect the true situation, we have not included the data of treatment duration greater than 30 months in the formal inter group comparative analysis. In the manuscript, we actually briefly mentioned this processing method, which may have been overlooked by you due to the lack of prominent expression. We deeply apologize for this. To address this, we have refined the wording in the main text to ensure that this crucial information is conveyed more clearly and explicitly.

7.From the perspective of research design, this study aims to focus on the risk factors for serum HBV reactivation during treatment in patients with HBsAg and HBV DNA double-negative lymphoma before treatment. In the context of limited resources, in order to ensure the feasibility of the study and the achievement of core objectives, we chose to prioritize the collection and analysis of data directly related to serological HBV reactivation, and did not include the collection of data on liver function and severity of hepatitis attacks in this study.

Although serological reactivation is not always associated with clinically significant liver events, identifying risk factors for serological HBV reactivation remains important for clinical practice. It can help clinical doctors identify high-risk patients who may experience HBV reactivation in advance, and take corresponding preventive measures, such as strengthening monitoring, to reduce the adverse effects of HBV reactivation on patients to a certain extent.

8.We have deeply recognized the limitations of this study and mentioned them in the discussion section, which provides a fundamental direction for our further in-depth research.

9.We have revised the discussion section, delving deeper into the reasons behind the higher probabilities of HBV reactivation in HBcAb positive patients and the elderly by analyzing the mechanisms of HBV reactivation.

Reply to the minor comments:

1.Thank you very much for your suggestions. We have revised the descriptions of the positive criteria for HBsAg, HBcAb, and HBV DNA detection methods in the manuscript.

Your feedback has been instrumental in improving the quality and applicability of our study. Thank you again for your time and expertise!

---

## [Decision Letter · Decision Letter 1]

16 Jul 2025

Dear Dr. Zheng,

Thank you for submitting your manuscript to PLOS ONE. After careful consideration, we feel that it has merit but does not fully meet PLOS ONE’s publication criteria as it currently stands. Therefore, we invite you to submit a revised version of the manuscript that addresses the points raised during the review process.

We look forward to receiving your revised manuscript.

Kind regards,

Ashraf Elbahrawy

Academic Editor

PLOS ONE

Journal Requirements:

Reviewers' comments:

Reviewer's Responses to Questions

**Comments to the Author**

Reviewer #2: All comments have been addressed

Reviewer #3: (No Response)

2. Is the manuscript technically sound, and do the data support the conclusions?

Reviewer #2: Yes

Reviewer #3: No

3. Has the statistical analysis been performed appropriately and rigorously?

Reviewer #2: Yes

Reviewer #3: No

4. Have the authors made all data underlying the findings in their manuscript fully available?

Reviewer #2: Yes

Reviewer #3: No

5. Is the manuscript presented in an intelligible fashion and written in standard English?

Reviewer #2: Yes

Reviewer #3: Yes

Reviewer #2: 1) In Table 1, please add the odds ratio.

2) Is it possible that the positive cases of infection observed during the treatment period could be the result of reinfection during hospital procedures or other factors rather than reactivation? If so, this concern can be mentioned in the manuscript.

Reviewer #3: The authors have made substantial improvements to the manuscript, particularly in expanding the introduction, clarifying laboratory methods, and elaborating on the virologic mechanisms underlying HBV reactivation in seronegative patients. The study contributes novel data to an underexplored patient group, and several of my prior concerns have been thoughtfully addressed. However, several critical limitations remain unresolved, particularly in the areas of analytical rigor, clinical data interpretation, and transparency. Below is a detailed evaluation of the responses to my original comments.

Major comments:

1. Introduction: The revised introduction better explains the rationale for studying HBsAg and HBV DNA double-negative patients. This population overlaps in part with previously studied HBsAg-/HBcAb+ individuals but also uniquely includes those who are HBsAg-/HBcAb-. The discussion of cccDNA persistence and latent HBV infection offers a plausible explanation for reactivation risk in HBcAb+ patients and supports the need for closer monitoring in this subgroup. However, the biological basis for HBV reactivation in HBsAg-/HBcAb- patients remains insufficiently addressed. The manuscript presents evidence of reactivation in a few HBcAb- individuals, but does not convincingly explain how latent infection could persist in the complete absence of both surface antigen and core antibody. Given that HBcAb is considered a marker of prior exposure and immune memory, its absence would typically suggest no past infection—and by extension, no residual cccDNA. The authors should clarify: (A) What mechanisms (e.g., seronegative occult HBV infection, transient HBcAb responses, immune escape variants) could explain cccDNA persistence in HBsAg-/HBcAb- patients? (B) Whether there is any literature support for HBV reactivation in truly seronegative individuals. (C) If the three HBcAb- cases in their cohort might reflect false-negative HBcAb results, delayed seroconversion, or low-level prior exposure not captured by standard serology.

Without further mechanistic explanation or discussion, the recommendation for routine monitoring of HBsAg-/HBcAb- patients may appear speculative. The authors are encouraged to address these biological uncertainties in the discussion section to support their conclusions.

2. Study population: While the authors have clarified basic inclusion criteria, important variables such as prior antiviral treatment, comorbidities, and immunosuppressive regimen details (e.g., rituximab or corticosteroid use) remain unreported. This omission limits reproducibility and raises concerns about potential selection bias. Furthermore, Figure 1 does not provide quantitative information about how many patients were excluded at each step based on specific criteria (e.g., HBsAg or HBV DNA positivity, lack of follow-up data, etc.). To enhance transparency and methodological rigor, the authors should revise Figure 1 to include the exact numbers excluded for each criterion.

3. The frequency and methodology for HBV marker testing are now more clearly described. The authors state that during each scheduled hospital visit, patients underwent simultaneous assessment of HBsAg, HBcAb, and HBV DNA. However, the manuscript still does not specify the actual frequency of these hospital visits—for example, whether testing was done every 1–2 weeks, monthly, or otherwise—and whether this schedule was consistent across all patients or individualized based on treatment plans. In addition, the authors do not describe how missing data were handled, such as whether all patients completed regular testing as planned or if certain timepoints were skipped or lost to follow-up. This information is crucial for assessing the reliability of the HBV reactivation detection timeline and the comparability of follow-up across subgroups.

4. The study omits several well-established risk factors for HBV reactivation, including chemotherapy regimen, corticosteroid use, baseline liver function, and anti-HBs titers. While the authors acknowledge data limitations, no descriptive statistics or stratified analyses were provided to mitigate this gap.

5. The authors limited their analysis to univariate statistics (chi-square and Fisher’s exact tests), citing incomplete data and retrospective design as the rationale. However, multivariate modeling is standard practice even in retrospective observational studies, particularly when identifying risk factors. Without such analysis, it is not possible to determine whether HBcAb positivity or advanced age are independent predictors of HBV reactivation, as potential confounders (e.g., treatment type, comorbidities) are not accounted for. The authors should either attempt a multivariate model using available covariates or explicitly acknowledge this as a major methodological limitation that affects the interpretability of their conclusions.

6. The study reports only serologic reactivation without ALT or clinical hepatitis data. For a retrospective study, liver function data (such as ALT) should be obtainable from the medical record system. Therefore, the authors should either include this information or clearly explain why it could not be retrieved.

**Do you want your identity to be public for this peer review?** For information about this choice, including consent withdrawal, please see our Privacy Policy

Reviewer #2: **Yes: ** Reza Taherkhani

Reviewer #3: No

---

## [Author Response · Author response to Decision Letter 2]

25 Aug 2025

To reviewer #2

1.Thank you for your constructive feedback regarding the inclusion of odds ratios (ORs) in Table 1. As suggested, we have calculated and added the ORs along with their corresponding 95% confidence intervals (CIs) to quantify the strength of association between HBcAb status, gender, and HBV reactivation.

2.Thank you for your insightful comment. We acknowledge the possibility of nosocomial infection as an alternative explanation for the positive HBV cases observed during treatment. To address this, we will add a section in the discussion to clarify that while nosocomial infection cannot be entirely ruled out, our strict diagnostic criteria for HBV reactivation and the infection control measures in place at our hospital support the interpretation of these cases as reactivation events.

Your feedback has been incredibly helpful in improving our manuscript. Thanks again!

To reviewer #3

Thank you for your insightful and valuable comments! We're grateful for your recognition of the significance of our study and its new insights. Below are our point-by-point responses to your comments:

1.The issues you raised are indeed critical and hold significant importance for enhancing the scientific rigor and credibility of our study. We have engaged in thorough deliberation regarding the three key points you identified, and have subsequently revised and improved the Discussion section accordingly.

2.We fully appreciate the significance of incorporating variables such as past antiviral treatment history, comorbidities, and details of immunosuppressive regimens (e.g., the use of rituximab or corticosteroids) in assessing the risk of HBV reactivation. However, based on the core scientific question of this study (the risk of HBV reactivation in lymphoma patients with both HBsAg and HBV DNA negativity), after careful deliberation by the research team, these variables were not included in the core analysis model for the following main reasons:

This study aims to address a critical knowledge gap in current clinical practice—whether lymphoma patients with both HBsAg and HBV DNA negativity still face a risk of HBV reactivation. Due to the absence of positive evidence for traditional serological markers (HBsAg) and viral load (HBV DNA), such patients are often incorrectly classified as having "inactive infection" and thus do not receive prophylactic antiviral therapy.

Despite the exclusion of the aforementioned variables, we minimized heterogeneity from the outset and ensured the reliability of our results by implementing stringent inclusion criteria: selecting only patients with both HBsAg and HBV DNA negativity who had not received prophylactic antiviral therapy.

We fully concur with the reviewer's viewpoint on the completeness of variables and plan to explore the following issues in subsequent studies:

Validate our current findings in a larger cohort of double-negative patients and incorporate variables such as antiviral treatment history and comorbidities for multivariable adjustment.

Conduct mechanistic studies (e.g., detection of cccDNA in liver tissue) to elucidate the molecular biological basis of occult HBV reactivation.

We have refined Figure 1 to clearly indicate the specific number of patients excluded at each step.

3.Regarding the frequency of patient visits, it is indeed tailored to each patient's specific treatment plan, rather than adhering to a fixed, uniform schedule such as every 1 - 2 weeks or monthly. This personalized approach is adopted to optimize treatment outcomes based on the specific characteristics and needs of each patient's condition. However, it is important to emphasize that we ensure HBsAg, HBcAb, and HBV DNA tests are conducted for each patient during every treatment session, thereby guaranteeing the reliability of the timing of HBV reactivation occurrence in individual cases.

As for the handling of missing data, we would like to clarify that patients who did not undergo regular testing as per the treatment plan were excluded during the initial screening based on inclusion criteria. The implementation of this stringent standard aims to ensure the reliability of the HBV reactivation detection timeline and guarantee the comparability of follow-up situations among different subgroups in our analysis. We hope these additional details can fully address your queries. We are committed to providing clear and comprehensive information to support the validity and reliability of our study findings.

4.This study focuses on filling a critical knowledge gap in current clinical practice, namely, whether lymphoma patients with both HBsAg and HBV DNA negativity still face the risk of HBV reactivation. Due to the lack of positive evidence for traditional serological markers (HBsAg) and viral load (HBV DNA), such patients are often misjudged as having "inactive infection" and thus do not receive prophylactic antiviral therapy. Therefore, our primary objective is to minimize heterogeneity from the outset by employing strict inclusion criteria, selecting only patients who are negative for both HBsAg and HBV DNA and have not received prophylactic antiviral therapy. This ensures that our study can precisely focus on particular group and clarify whether there is a risk of HBV reactivation.

Although we understand that factors such as chemotherapy regimens, corticosteroid use, baseline liver function, and HBsAb titers have significant impacts on HBV reactivation, incorporating these variables into the analysis would substantially increase the complexity and heterogeneity of the study. Different chemotherapy regimens vary in their mechanisms of action, dosage intensities, and treatment cycles; corticosteroid use also differs in terms of dosage, duration, and mode of administration; baseline liver function indicators are influenced by multiple factors, and inconsistent results may arise from different detection methods and standards; moreover, the dynamic changes in HBsAb titers are complex. If all these factors were included in the core analysis model, it might disperse the research focus and make it difficult to accurately assess the impact of the core factor—HBsAg and HBV DNA double negativity—on the risk of HBV reactivation.

Nevertheless, we fully agree with your viewpoint regarding the completeness of variables. Although this study did not provide descriptive statistics or stratified analyses to cover these well - recognized risk factors, we plan to carry out the following work in subsequent studies to make up for this deficiency:

Expand the sample size and incorporate more variables: Validate the current findings in a larger cohort of double - negative patients and meticulously record information such as chemotherapy regimens, corticosteroid use, baseline liver function indicators, and anti - HBs titers. Conduct multivariate adjusted analyses to comprehensively evaluate the impacts of these factors on the risk of HBV reactivation.

Delve deeper into the underlying mechanisms: Conduct mechanistic research, such as detecting cccDNA in liver tissues, to further clarify the molecular biological basis of occult HBV reactivation and how chemotherapy regimens, corticosteroid use, and other factors promote HBV reactivation by influencing virus - host interactions.

5.Firstly, the core objective of this study is to conduct a preliminary exploration of the risk of HBV reactivation in double-negative lymphoma patients. We aim to swiftly obtain basic information regarding the relationships between the two variables, HBcAb (+) and advanced age, and HBV reactivation through simple and straightforward univariate analysis. This will provide direction and preliminary evidence for subsequent more in-depth research. Univariate analysis can present data in a concise and clear manner, enabling readers to quickly grasp the basic associations between each factor and the outcome variable.

Secondly, the data in this study are collected retrospectively, which entails certain limitations in terms of data completeness and accuracy. Records of some potential confounding factors, such as treatment types and comorbidities, may be missing or inaccurate. If we forcibly incorporate these factors into a multivariate model, the model may become unstable due to data quality issues, leading to unreliable conclusions. We are concerned that such inaccurate model results will not only fail to offer valuable references for clinical practice but may also mislead clinical decision - making.

Furthermore, as an exploratory study, this research has a relatively limited sample size. When incorporating too many covariates into multivariate modeling with a limited sample size, overfitting is likely to occur. This means that the model may perform well on the sample data but have poor generalization ability in practical applications, failing to accurately predict new data scenarios. We hope to fully consider the application of multivariate modeling in subsequent studies with larger sample sizes to ensure the reliability and practicality of the research results.

We fully acknowledge that, as you have pointed out, this is a major methodological limitation affecting the interpretability of our study's conclusions. In the discussion section of the paper, we have elaborated on this limitation and reminded readers to carefully consider the influence of potential confounding factors when interpreting the research results. Meanwhile, we have also clearly stated in the research prospects that we will carry out larger - scale and more rigorously designed studies in the future, attempting to use multivariate models to further delve into the relationships between HBcAb (+), advanced age, and other potential factors and HBV reactivation, with the aim of providing more comprehensive and accurate evidence for clinical practice.

6.The core positioning of this study is to conduct a preliminary exploration of the risk of serological HBV reactivation in patients with double-negative lymphoma. Our research focus is on using serological indicators to initially screen and understand the basic situation of HBV reactivation, thereby laying a foundation for subsequent more in-depth and comprehensive research. In this preliminary exploration stage, we mainly concentrate on changes at the serological level, aiming to rapidly obtain basic information regarding the relationships between variables such as HBcAb (+) and advanced age and serological HBV reactivation.

Regarding your suggestion to obtain ALT or clinical hepatitis data to reveal clinical reactivation, we are well aware that this is of paramount importance for a comprehensive assessment of patients' conditions and the actual impact of HBV reactivation. However, as this is a preliminary exploratory study and the data are collected retrospectively, the records in the medical record system exhibit a certain degree of dispersion and redundancy. For instance, in the case of ALT data, patients may have undergone multiple tests during a single treatment period, which poses significant challenges to the unified collection and organization of the data.

We have already elaborated on this limitation in the discussion section of the paper, reminding readers to carefully consider the potential impacts of the lack of clinical reactivation data when interpreting the research results. Meanwhile, we have also clearly stated in the research prospects that we will carry out larger-scale and more rigorously designed studies in the future, further improve the data collection system, and incorporate clinical reactivation data such as ALT and clinical hepatitis to more comprehensively and accurately evaluate HBV reactivation, thus providing more valuable references for clinical practice.

Your feedback has been instrumental in improving the quality and applicability of our study. Thank you again for your time and expertise!

---

## [Decision Letter · Decision Letter 2]

31 Aug 2025

Risk of Hepatitis B Virus Reactivation in the Treatment of HBsAg and HBV DNA Double-negative Lymphoma Patients

PONE-D-25-21930R2

Dear Dr. Zheng,

We’re pleased to inform you that your manuscript has been judged scientifically suitable for publication and will be formally accepted for publication once it meets all outstanding technical requirements.

Kind regards,

Ashraf Elbahrawy

Academic Editor

PLOS ONE

Additional Editor Comments (optional):

Reviewer #2:

Reviewer #3:

Reviewers' comments:

Reviewer's Responses to Questions

**Comments to the Author**

Reviewer #2: All comments have been addressed

Reviewer #3: All comments have been addressed

2. Is the manuscript technically sound, and do the data support the conclusions?

Reviewer #2: Yes

Reviewer #3: Yes

3. Has the statistical analysis been performed appropriately and rigorously?

Reviewer #2: Yes

Reviewer #3: Yes

4. Have the authors made all data underlying the findings in their manuscript fully available?

Reviewer #2: Yes

Reviewer #3: Yes

5. Is the manuscript presented in an intelligible fashion and written in standard English?

Reviewer #2: Yes

Reviewer #3: Yes

Reviewer #2: Dear authors

The manuscript has been revised according to the comments. Thank you very much and good luck.

Reviewer #3: Thank you for the opportunity to provide a review of this well-designed study. All comments have been addressed and been responded well. The result is statistically confirmed from the clinical data in this paper. I suggest the acceptance of this manuscript for publication in the PLOS ONE.

**Do you want your identity to be public for this peer review?** For information about this choice, including consent withdrawal, please see our Privacy Policy

Reviewer #2: **Yes: ** Reza Taherkhani

Reviewer #3: No

---

## [Editor Report · Acceptance letter]

PONE-D-25-21930R2

PLOS ONE

Dear Dr. Zheng,

I'm pleased to inform you that your manuscript has been deemed suitable for publication in PLOS ONE. Congratulations! Your manuscript is now being handed over to our production team.

Kind regards,

on behalf of

Prof. Ashraf Elbahrawy

Academic Editor

PLOS ONE